# Metastatic Tumors of the Oro-Facial Tissues: Clear Cell Renal Cell Carcinoma. A Clinico-Pathological and Immunohistochemical Study of Seven Cases

**DOI:** 10.3390/jcm9041151

**Published:** 2020-04-17

**Authors:** Saverio Capodiferro, Luisa Limongelli, Mauro Giuseppe Mastropasqua, Gianfranco Favia, Carlo Lajolo, Giuseppe Colella, Angela Tempesta, Eugenio Maiorano

**Affiliations:** 1Complex Operating Unit of Odontostomatology, Interdisciplinary Department of Medicine, Università degli Studi di Bari Aldo Moro, 70100 Bari, Italy; luisanna.limongelli@gmail.com (L.L.); gianfranco.favia@uniba.it (G.F.); angelatempesta1989@gmail.com (A.T.); 2Anatomic Pathology, Department of Emergency and Organ Transplantation, Università degli Studi di Bari Aldo Moro, 70100 Bari, Italy; mauro.mastropasqua@gmail.com (M.G.M.); eugenio.maiorano@uniba.it (E.M.); 3Head and Neck Department, “Fondazione Policlinico Universitario A. Gemelli-IRCCS”, School of Dentistry, Università Cattolica del Sacro Cuore, 00100 Rome, Italy; carlo.lajolo@unicatt.it; 4Policlinico I, Università Luigi Vanvitelli, 80100 Naples, Italy; giuseppe.colella@unicampania.it

**Keywords:** oral tumors, jawbones, salivary glands, metastatic tumors, clear cell renal cell carcinoma

## Abstract

Metastases to orofacial tissues are infrequent, their incidence being 1%–8% of malignant oral tumors, sometimes manifesting as the first clinical sign of an occult cancer. Renal cell carcinoma (RCC) is the second most common metastatic carcinoma to the oro-facial tissues, involving the jawbones, gingiva, oral mucosa, tongue or salivary glands. Also, RCC frequently displays a prominent clear cell component, which may predominate in the clear cell renal cell carcinoma subtype (CCRCC) and histologically mimic many other clear cell tumors, both benign and malignant, which can be epithelial (from keratinizing epithelia, cutaneous adnexa, salivary glands and odontogenic epithelium), melanocytic or mesenchymal in origin. In view of the necessity for prompt and accurate diagnosis of such unusual neoplasms, we report on the salient clinico-pathological features of 7 CCRCC metastatic to the oro-facial tissues, and highlight their immunohistochemical profile, to more accurately discriminate this neoplasm from other tumors of the oral cavity with a prominent clear cell component.

## 1. Introduction

Metastatic tumors involving the oro-facial tissues are infrequent, their incidence ranging between 1% and 8% of all oral malignant tumors [1,2,3,4]. With the exception of malignant tumors of childhood, the peak incidence is in the 5th–7th decades [2]. They can be the first sign of an occult cancer or manifest during the clinical follow-up of a patient with a known primary carcinoma [5,6,7].

Metastases to the oro-facial tissues can involve the oral mucosa, jawbones or the salivary glands, the most frequent primary localization being lung, kidney, prostate and colon-rectum in males, uterus, breast, lung and ovary in females [1,2,8,9].

The predilection of metastatic neoplasms for specific sites in the oro-facial region may be influenced by peculiar clinical conditions, such as the gingival-parodontal soft tissues of dentates with inflammatory lesions of the parodontum, or the same tissues in edentulous individuals bearing prostheses. In such instances, the re-organization of the local blood flow, following inflammation or induced by the pressure of the prosthesis, have been postulated to facilitate the metastatic growth [10].

The jawbones and, in particular, the molar and premolar regions of the mandible and maxilla frequently are involved in view of their rich vascularization and high bone marrow content. Also, metastases may develop at post-extraction sites, possibly as the consequence of increased blood flow following organization of the blood clot [1,2,8,10].

Clear cell renal cell carcinoma (CCRCC) represents 70% of all malignant renal tumors [11,12]; it frequently metastasizes through the blood stream, particularly to the lungs, bones, liver, brain and to the opposite kidney [11,13]. Metastatic CCRCC to the oro-facial tissues have been occasionally reported [1,2,11,14] and, in some instances, they represented the initial manifestation of the disease [2,6,7,13] Due to high glycogen and lipid content, the tumor cells of CCRCC display evident cytoplasmic vacuolization and clearing and may mimic other neoplasms of odontogenic or salivary gland origin that more commonly affect this area. [2,11,12,14,15,16,17].

Consequently, the oral localization of an occult CCRCC surely may represent a diagnostic challenge, especially when the clinical work up is limited to the cervico-facial region [6,7,18,19,20,21].

Moreover, although CCRCC shows peculiar morphologic features, other renal tumors with similar morphology must be taken into account, namely clear cell papillary renal cell carcinoma (CCPRCC) even if no metastases of CCPRCC to the oro-facial tissues have been reported so far, possibly due to its indolent behavior [14]. 

The aim of this study was to extensively review the clinico-pathological features of 7 CCRCC metastatic to the oro-facial tissues, to better define their differential diagnostic features in comparison with other clear cell tumors of the same sites. For this purpose, a short series of different salivary gland, and odontogenic tumors showing prominent clear cell features was included in the study.

## 2. Materials and Methods

The clinical charts of all patients with secondary neoplasms of the oro-facial tissues observed at the Interdisciplinary Department of Medicine, Section of Dental Sciences and Surgery of the University of Bari during the period 1971–2000 were collected. Among these, 7 cases of CCRCC with histopathological evaluation of both the primary and the metastatic tumors were identified and included in this study.

In addition, the clear cell variants of several tumors of the salivary glands (acinic cell carcinoma—3 cases, adenoid cystic carcinoma—5 cases, epithelial-myoepithelial carcinoma—5 cases, hyalinizing clear cell carcinoma—2 cases, mucoepidermoid carcinoma—5 cases, myoepithelioma—2 cases, oncocytoma—2 cases, pleomorphic adenoma—5 cases) or odontogenic origin (ameloblastoma—5 cases and clear cell odontogenic carcinoma—2 cases), showing typical histopathological features were included for comparative purposes of the histochemical and immunohistochemical characteristics.

The surgical samples from all neoplasms had been formalin-fixed, paraffin-embedded and stained with hematoxylin-eosin (H&E). Additional sections were cut from the paraffin blocks and stained with periodic acid-Schiff (PAS), with and without diastase treatment, Mucicarmin and Gomori’s reticulin. Consecutive sections were collected on positively-charged slides and used for the immunohistochemical stains that were carried out with modified avidin-biotin peroxidase technique using an automated immunostainer (Autostainer, Agilent Technologies, Glostrup, Denmark). The primary antibodies employed for the study are listed in Table 1, along with the appropriate positive controls. Negative controls were obtained by substituting the primary antibodies with pre-immune rabbit or mouse sera.

This study was carried out in accordance with the code of ethics of the world medical association (Declaration of Helsinki) and approved by internal ethical committee (study number 4575, prot. 1442/C.e). Patients released informed consent on diagnostic and therapeutic procedures and for the possible use of biological samples for research purposes.

## 3. Results

### 3.1. Clinical Features

The salient clinical-pathological features of the 7 CCRCCs included in this study are illustrated in Table 2. Overall, there were 5 males and 2 females (mean age: 57 years, range: 45–69) and in 2 cases the oral metastatic deposits were the first clinical manifestations of the renal tumors. In one such cases and in two others, additional synchronous metastases were radiologically detected in the lungs, thyroid, skin of a finger and in a vertebra.

As to the oral-facial localizations, the mandible was involved in 3 cases (molar, incisor and condilar region, respectively) (Case 4, Figure 1a,b), the parotid gland in 2, the remaining 2 cases being localized to the anterior maxillary gingiva and to the tongue (Case 2, Figure 2). The mandibular metastases radiologically appeared as radiolucent lesions, with ill-defined borders and cortical erosion. In addition, one of them also expanded into the adjacent soft tissues, thus causing submucosal swelling on the lingual aspect of the gingiva.

The intraoral metastatic CCRCCs (Cases 1 and 2) presented as large fungating masses of purplish red to white-reddish color and showed ulceration of surface epithelium. The one localized to the maxillary gingiva also involved the palatal mucosa and, radiologically, was associated with bone rarefaction of the incisor maxillary bone.

The parotideal metastatic neoplasms appeared as rapidly growing intra-glandular expansions of large dimensions.

### 3.2. Histopathology

All CCRCCs showed either solid or lobular architecture and were composed by large clusters of polyhedral cells with prominent cytoplasmic clearing and rather indistinct borders (Figure 3 and Figure 4); the tumor cells possessed rounded to oval nuclei, with occasional prominent nucleoli. Intra-tumoral hemorrhage was a constant feature of these neoplasms and was detectable either as small intercellular aggregates of red blood cells or as large hemorrhagic areas that displaced the adjacent carcinomatous cells.

Also, all tumors showed rich intra-lesional vascularization, mainly in the form of sinusoidal vessels composed by a thin basal lamina, covered by flattened endothelial cells. Quite frequently, the sinusoids were enlarged and contained collections of erythrocytes. The typical vascularity of CCRCCs was better highlighted after Gomori’s reticulin stain and was not observed in any other clear cell neoplasm included in this series for comparison, which only showed small-sized intra-tumoral capillary vessels.

All CCRCCs demonstrated typical PAS-positive intracytoplasmic granules that disappeared after diastase treatment, consistent with glycogen accumulation. The concentration of such PAS-positive granules was quite variable from case to case, two of these being entirely composed by PAS-positive cells, the remainder showing large clusters or only scattered glycogen-containing cells.

### 3.3. Immunohistochemistry

Overall, all CCRCCs were consistently positive for cytokeratins AE1/AE3, Epithelial Membrane Antigen (EMA) (Figure 5a), PAX-8 (Figure 5b), Renal Cell Carcinoma Antigen (RCCAg) and vimentin (Figure 5c) and also showed at least focal immunoreactivity for Mitochondria and S-100 protein, while 5/7 cases were immunoreactive for CD10 (Figure 5d), as detailed in Table 3. Differently, no CCRCC was immunoreactive for muscle markers (smooth muscle actin, calponin, myosin), calretinin, cytokeratin 7, CD117 and Glial Fibrillary Acidic Protein (GFAP). Comparatively, salivary gland tumors immunoreactivity was distributed as follows:cytokeratins AE1/AE3, cytokeratin 7 and EMA were detected in all tumors, with less extensive positivity for cytokeratins AE1/AE3 in myoepithelioma, for cytokeratin 7 in epithelial-myoepithelial carcinoma (EMEC), hyalinizing clear cell carcinoma (HCCC) and myoepithelioma, and for EMA in HCCC and myoepithelioma;muscle markers (smooth muscle actin, calponin, myosin) were highlighted in adenoid cystic carcinoma, EMEC, myoepithelioma and pleomorphic adenoma;CD117 was extensively positive in adenoid cystic carcinoma and EMEC and, to a lesser extent, in mucoepidermoid carcinoma and pleomorphic adenoma;GFAP was diffusely expressed by pleomorphic adenoma and only focally detectable in EMEC and myoepithelioma;Mitochondria Ag was consistently expressed by oncocytoma and occasionally by acinic cell carcinoma and adenoid cystic carcinoma;S-100 protein and vimentin were present in adenoid cystic carcinoma, EMEC, myoepithelioma and pleomorphic adenoma;Calretinin positivity was highlighted in a minority of tumor cells of acinic cell carcinoma;CD10, PAX8 and RCCAg were consistently negative in all tumors of this subgroup;as to odontogenic tumors:cytokeratins AE1/AE3, cytokeratin 7 and EMA were detected in odontogenic carcinoma but cytokeratins AE1/AE3 only was present in ameloblastoma;calretinin was unequivocally present in ameloblastoma but absent in odontogenic carcinoma, as was vimentin, which decorated a small number of tumor cells;all other antigens were invariably lacking in all odontogenic tumors.

## 4. Discussion

Distant metastases from RCC are very common and usually multiple to different organs, such as lungs (50%–60%), bones and liver (30%–40%), and head and neck (12%–16%) [1,2]. Among the latter, 50% of the metastases were detected in the thyroid, nose and paranasal sinuses, and pharynx [11,13,20], at variance with solitary metastases to the head and neck, which are exceedingly rare, accounting for only 1% of the cases [2,11,19,20].

Distant metastases to the oro-facial tissues are very rare [2] and may involve the jaws [5,8,10] especially the mandible, or the soft tissues, mostly gingiva [2,5,9] and, more rarely, the tongue, with a prevalence of 0.17% [7,15,22,23].

According to histopathologic subtypes, adenocarcinoma, renal cell carcinoma and squamous cell carcinoma are the most common metastatic tumors, usually detected in the fifth and sixth decade of life, with slight predominance in males [1,2,3,19,24]. Several single case reports and small series of metastases in the head and neck region are available in the literature [2,3,4,6,8,9,10,25].

The histologic detection of a prominent clear cell component in head and neck neoplasms should always lead one to consider a possible secondary localization of an unknown neoplasm in the differential diagnosis, the latter being a challenging situation both for the identification of the lesion as a metastatic neoplasm, and for the identification of the site of the primary tumor. Renal cell tumors show different morphology according to their histotype so that, quite often, an immunohistochemical evaluation is deemed necessary for a correct histological diagnosis, and this is obviously more evident in the case of metastases. Although there is not a unique marker which can be considered specific for renal origin, distinct combinations of antibodies may facilitate the identification of the kidney as the site of origin of the tumor. Currently, PAX8, although expressed by other epithelial tumors, seems the most useful antibody at this regard, exceeding classical older markers, such as CD10 or RCCAg, an antibody directed against a brush border protein. Carbonic anhydrase IX (CAIX) also is expressed in renal tumors, mostly CCRCC, and stains the membranes of tumor cells. CAIX also is expressed in CCPRCCs, but it shows a so-called cup-like staining pattern, decorating the basal and the lateral borders of cells, while the luminal aspect remains unstained. Furthermore, CCPRCCs shows diffuse cytokeratin 7 immunostaining, at variance with CCRCC which is cytokeratin 7 negative. In addition, PAX2 immunostaining reproduces the staining pattern of PAX8, the latter being more sensitive. 

Therefore, in consideration of the reported typical reactivity of CCRCC, PAS, with and without diastase treatment, vimentin, CD 10, EMA, RCCAg, and PAX-8 should be included [15,26,27,28,29]. In consideration of the results of the current study, we argue the practical usefulness of EMA and vimentin in this setting, due to consistent immunoreactivity for such markers in many distinct salivary gland tumors and, to a lesser extent, in odontogenic tumors as well.

Furthermore, though for speculative purposes only, clear cell sarcoma of the kidney should be considered in the differential diagnosis, at least in pediatric patients, being the second most common malignant tumor in children. Unfortunately, no specific immunohistochemical marker is available for this tumor type, with the exception of cyclin D1, which seems to distinctly and selectively decorate this sarcoma [30].

When occurring in major salivary glands, the differential diagnosis of clear cell neoplasms includes mucoepidermoid carcinoma (MEC). The absence of micro-vessels surrounding the nests of clear cells, and the presence of peripherally displaced nuclei, help to differentiate MEC from CCRCC, along with the co-expression of vimentin and CD10 and the absence of reactivity for mucicarmine in the latter. In this context, PAS consistently stains both CCRCC and MEC but only the latter remains positive following diastase treatment, due to mucin and not glycogen intra-cytoplasmic content [11,19,20,21].

Such findings are useful to differentiate CCRCC from MEC with intraosseous localization to the jawbones and with oral mucosal localization, in order to exclude a possible origin from minor salivary glands. 

Besides MEC, other salivary gland tumors, such as epithelial-myoepithelial carcinoma, oncocytomas, hyalinizing clear cell carcinoma (HCCC) and acinic cell carcinoma (ACC) may display a clear cell component [11,12,16,17]. However, the presence of a clear cell component is not the exclusive morphologic feature characterizing such tumor types, and a correct diagnosis may usually be achieved based on to their typical histopathological features [11]. With regards to ACC, it should be underlined that PAS-positive, diastase-resistant zymogen granules are evident in this tumor, along with positive immunostaining for S-100 and CD117, and such peculiarities surely facilitate the distinction from CCRCC [11,20]. HCCC, instead, is a tumor predominantly occurring in minor salivary glands, which is immunostochemically negative for S100, smooth muscle actin, calponin and glial fibrillary acidic protein (GFAP) but positive for pan-cytokeratins and p63. The latter marker may be of pivotal relevance in this regard, being consistently negative in renal cell carcinoma [2,11,13,26]. 

The differential diagnosis of jawbone metastases also includes clear cell ameloblastoma (CCA), calcifying epithelial odontogenic tumor (CEOT) and clear cell odontogenic carcinoma (CCOC) [8,10,12,18]. The presence of stellate reticulum-like cells and of peripheral palizading of columnar tumor cells are useful to rule out CCA, the latter being also immunoreactive for calretinin, at variance with CCRCC. CEOT can show a clear cell component but this is usually admixed with eosinophilic cells with large nuclei and prominent nucleoli. The presence of calcified deposits and amorphous eosinophilic material positive for Congo red and birefringent under polarized light surely facilitates the diagnosis of CEOT, the latter being also PAS-negative, at variance with CCRCC. CCOC is characterized by clusters of clear cell nests with ameloblastoma-like peripheral palizading; the tumor cells usually are PAS-positive but, at variance with CCRCC, they demonstrate S-100 protein and melanoma-associated antigen (MAA) immunoreactivity [2,12,13,26].

In conclusion, metastatic CCRCC to the head and neck may be suspected, based on several morphological features; nevertheless, in view of the many different tumor types entering into the differential diagnosis, morphology must be supported by immunohistochemical stains for proper identification of the specific tumor type. Several markers may help in this regard, such as CD10 and RCCAg, which have been effectively used in this regard for many years. Furthermore, we would suggest incorporating PAX8 in the panel, which, although non-specific for the renal origin (being positive in endometrial, ovarian and thyroid cancers), has been proven useful in primary and metastatic RCCs. Recent studies, in fact, have demonstrated PAX8 positivity in up to 90% of metastatic CCRCCs [13,27,28,29,31,32]. For practical purposes, we would also like to emphasize the consistent lack of immuno-expression of muscle markers (e.g., smooth muscle actin, calponin and smooth muscle myosin heavy chain) in CCRCC, at variance with many primary tumors arising in the head and neck, specifically salivary gland tumors.

## Figures and Tables

**Figure 1 jcm-09-01151-f001:**
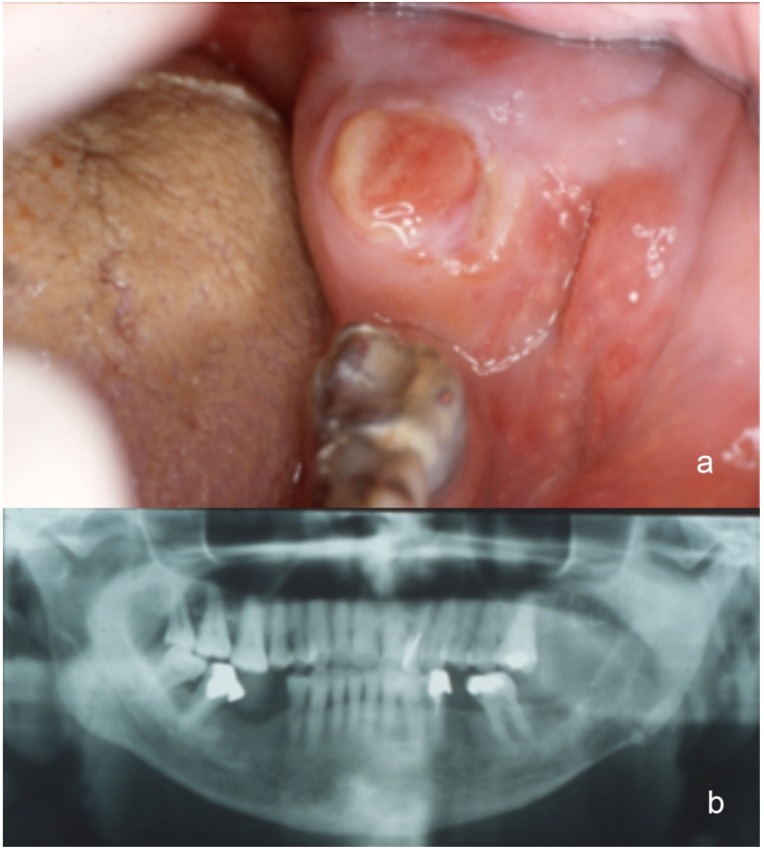
(**a**) Intra-oral presentation of a large and ulcerated mass in the retro-molar area, (**b**) showing an osteolytic appearance on panoramic radiograph, with extensive destruction of the cortical plates.

**Figure 2 jcm-09-01151-f002:**
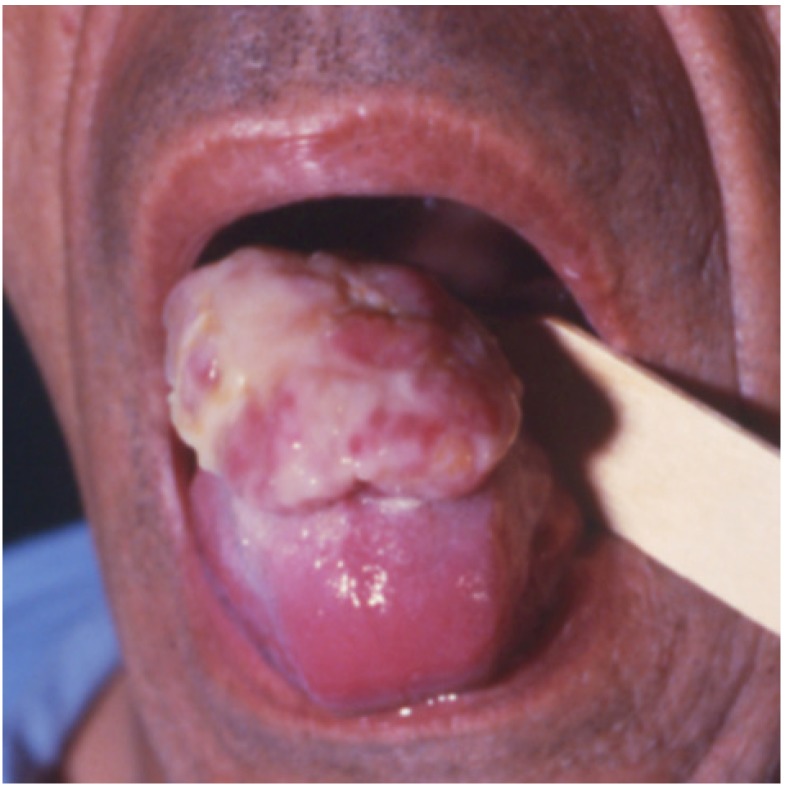
Large fungating mass affecting the middle portion of the tongue, with variegated white-yellowish to red discoloration.

**Figure 3 jcm-09-01151-f003:**
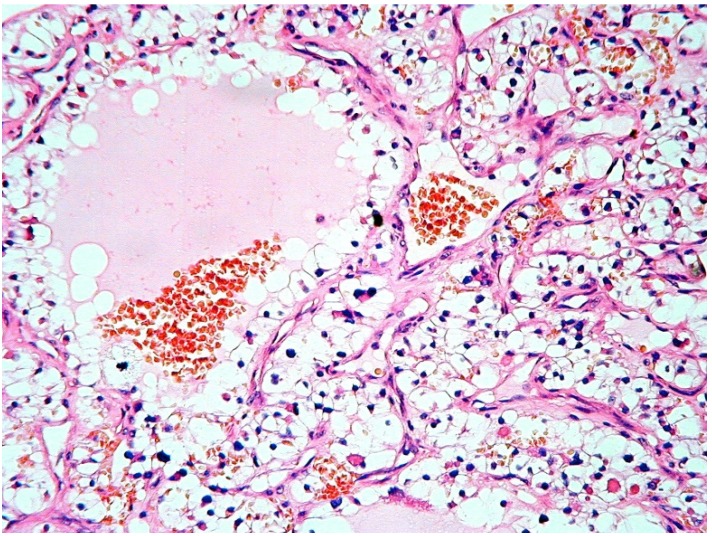
Morphologic features of classic clear cell renal cell carcinoma: the tumor cells are aggregated in solid clusters or in follicular-like structures and show extensive cytoplasmic clearing. Also, clusters of red blood cells and capillary vessels are easily detected among tumor cells. (Hematoxylin-eosin, original magnification ×200).

**Figure 4 jcm-09-01151-f004:**
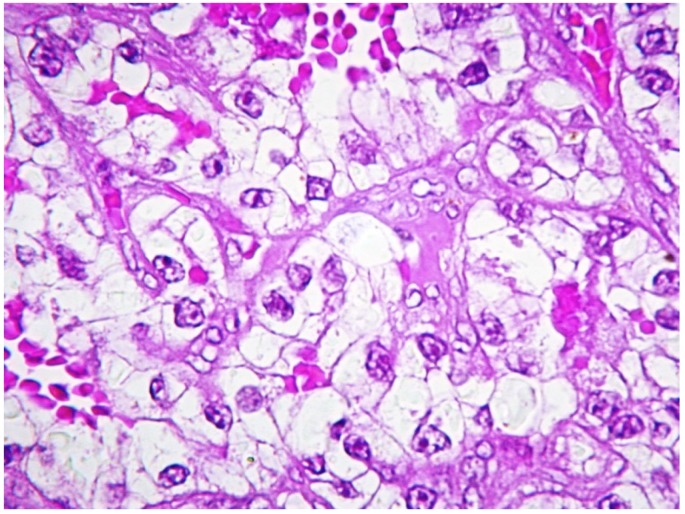
At higher magnification, the tumor cells display evident cytoplasmic clearing with scattered and poorly defined granules. The nuclei appear rounded to ovoid, with more or less prominent nucleoli. (Hematoxylin-eosin, original magnification ×400).

**Figure 5 jcm-09-01151-f005:**
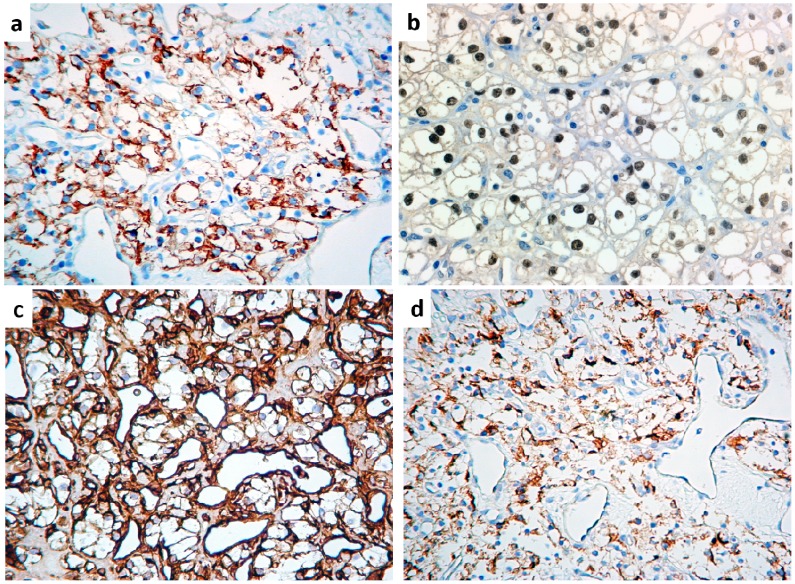
Immunohistochemical reactivity in clear cell renal cell carcinoma: the tumor cells display consistent positivity for (**a**) Epithelial Membrane Antigen, (**b**) PAX8, (**c**) Renal Cell Carcinoma Antigen and (**d**) CD10, allowing proper discrimination from salivary gland and odontogenic tumors. (Original magnification ×200).

**Table 1 jcm-09-01151-t001:** Pertinent data of the monoclonal antibodies used to immuno-characterize metastatic clear cell renal cell carcinoma.

Antigen	Clone	Dilution	Source	Positive control
Actin (smooth muscle)	1A4	1/200	Agilent Technologies (Santa Clara, CA, USA)	Leiomyoma
Calponin	CALP	1/400	Agilent Technologies	Leiomyoma
Calretinin	Cal6	1/50	Novocastra (Newcastle upon Tyne, UK)	Mesothelioma
CD 10	56C6	1/100	Novocastra	Lymph node
CD 117	polyclonal	1/100	Agilent Technologies	GIST
Cytokeratins (pan)	AE1/AE3	1/300	Agilent Technologies	Breast carcinoma
Cytokeratin 7	OV-TL 12/30	1/250	Agilent Technologies	Breast carcinoma
Epithelial membrane antigen (EMA)	E29	1/20	Agilent Technologies	Colon
Glial fibrillary acidic protein (GFAP)	6F2	1/100	Agilent Technologies	Brain
Mitochondria Ag	113-1	1/300	Menarini Diagnostics (Firenze, IT, USA)	Salivary oncocytoma
Myosin (SMMHC)	SMMS-1	1/50	Agilent Technologies	Leiomyoma
PAX-8	polyclonal	1/200	Bio Care Medical (The Hague, NL, USA)	Kidney
Renal cell carcinoma antigen (RCC Ag)	SPM314	1/100	Agilent Technologies	Kidney
S-100 Protein	polyclonal	1/50	Agilent Technologies	Brain
Vimentin	V9	1/50	Agilent Technologies	Leiomyoma

**Table 2 jcm-09-01151-t002:** Clinico-pathological features of 7 cases of clear cell renal cell carcinoma metastatic to the oro-facial tissues.

Case	Sex	Age	Site	Size	First Sign of Disease	Additional Metastases
1	F	69	Gingiva	5 cm	No	Lungs
2	M	56	Tongue	4 cm	No	Thyroid, opposite kidney, finger
3	M	45	Mandible	5.5 cm	No	-
4	M	63	Mandible	5 cm	Yes	Vertebra
5	M	55	Parotid Gland	2.2	Yes	-
6	F	55	Parotid Gland	1.8	No	-
7	M	60	Mandible	3.5 cm	No	-

**Table 3 jcm-09-01151-t003:** Immunohistochemical profile of tumors with prominent clear cell features that most frequently affect the oro-facial region.

	Actin (Smooth Muscle)	Calponin	Calretinin	CD 10	CD 117	CK AE1/AE3	CK 7	EMA	GFAP	Myosin (SMMHC)	Mitochondria	PAX8	RCCAg	S-100Protein	Vimentin
CCRCC	-	-	-	+	-	+	-	+	-	-	±	+	+	±	+
Acinic cell carcinoma	-	-	±	-	-	+	+	+	-	-	±	-	-	-	-
Adenoid cystic carcinoma	+	+	-	-	+	+	+	+		+	-	-	-	+	+
EMEC	+	+	-	-	+	+	±	+	±	+	-	-	-	+	+
HCCC	-	-	-	-	-	+	±	±	-	-	-	-	-	-	-
Mucoepidermoid carcinoma	-	-	-	-	±	+	+	+	-	-	-	-	-	-	-
Myoepithelioma	+	+	-	-	-	±	±	±	±	+	-	-	-	+	+
Oncocytoma	-	-	-	-	-	+	-	+	-	-	+	-	-	-	-
Pleomorphic adenoma	+	+	-	-	±	+	+	+	+	+	-	-	-	+	+
Ameloblastoma	-	-	+	-	-	+	-	-	-	-	-	-	-	-	+
Odontogenic carcinoma	-	-	-	-	-	+	+	+	-	-	-	-	-	-	-

Abbreviations: CCRCC = clear cell renal cell carcinoma; EMEC = epithelial-myoepithelial carcinoma; HCCC = hyalinizing clear cell carcinoma. CK = cytokeratins; EMA = epithelial membrane antigen; GFAP = glial fibrillary acidic protein; RCCAg = renal cell carcinoma antigen.

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
