# Peer review of "Metastatic Tumors of the Oro-Facial Tissues: Clear Cell Renal Cell Carcinoma. A Clinico-Pathological and Immunohistochemical Study of Seven Cases"

_jcm, 2020, doi:10.3390/jcm9041151_

Round 1

Reviewer 1 Report

Dear authors

Thank you for the submission of your study. However, several very minor issues have to be addressed:

  1. The sentence "…first clinical sign of an occult cancer" appears twice (S.18 and S.21), please omit one of them.
  2. Although the "kidney" is the second most common source of metastasis to the oral cavity and jawbones, by no means CCRCC is the second most common tumor to send metastasis to the oral cavity and jawbones (S.19), RCC in general – yes, as you wrote in the discussion (see for example Hirshberg et al. 2014).
  3. Please keep in mind that the "oral cavity" (S.19) does not include the jawbones and salivary glands.
  4. Figure 1b: Is there a pathologic fracture of the left mandible?
  5. Is the source of Figures 3 and 4 the oral cavity/jawbones? If not, please replace these figures with relevant figures.
  6. Figures' legend should not contain abbreviations.
  7. References list should be formatted according to the Journal's style.  

Author Response

Please find herein our point-to-point response. Thank you very much for suggestions

  1. The sentence "…first clinical sign of an occult cancer" appears twice (S.18 and S.21), please omit one of them.

R: The duplication has been corrected by omitting the second sentence alluding to metastases as the first sign of disease (Pag. 1, line 21)

  1. Although the "kidney" is the second most common source of metastasis to the oral cavity and jawbones, by no means CCRCC is the second most common tumor to send metastasis to the oral cavity and jawbones (S.19), RCC in general – yes, as you wrote in the discussion (see for example Hirshberg et al. 2014).

R: The sentence in the abstract has been re-formulated by substituting CCRCC with RCC (Pag. 1, lines 20-21)

  1. Please keep in mind that the "oral cavity" (S.19) does not include the jawbones and salivary glands.

R: “Oral cavity” has been replaced with “oro-facial tissues” when the jawbones and salivary glands were considered among the possibly affected sites (Pag. 1, line 19)

  1. Figure 1b: Is there a pathologic fracture of the left mandible?

R: A pathologic fracture is present in the left mandible and referred to in the corresponding figure legend as “extensive destruction of the cortical plates” (Pag. 4, line 8)

  1. Is the source of Figures 3 and 4 the oral cavity/jawbones? If not, please replace these figures with relevant figures.

R: Figures 3 and 4 were taken from cases 3 and 5 (table 2), corresponding to CCRCC localizations to the mandible and the parotid gland (Pag. 5)

  1. Figures' legend should not contain abbreviations.

All abbreviations have been removed from the legend of Fig. 5 (Pag. 8, lines 2-4)

  1. References list should be formatted according to the Journal's style.

R: The reference list has been formatted according to the instructions for authors

Reviewer 2 Report

This manuscript is well-written and provides a reasonable differential and discussion regarding the occasion when a clear cell neoplasm is encountered in the jaws. The authors have endeavored to help the reader distinguish amongst the various histopathologic possibilities in this regard, and the suggested histochemical and immunohistochemical studies are appropriate.

Should the manuscript move forward with publication, the authors should revise the piece referring to the tumor of interest as "clear cell renal cell carcinoma" rather than "cell clear" as the word-order "cell clear" does not follow convention. Minor editing and spell-check is requisite including:

line 46 "cloth" should be "clot"

line 60 should begin "The aim . . .,"

discussion: paragraph 4 line 4 should read "site of primary tumor"

discussion: spelling corrected to read "birefringent"

Finally, the manuscript would be significantly enhanced by the inclusion of a table that endeavors to comprehensively list each of the clear cell lesions one may encounter in the head and neck together with the relevant histochemical and immunohistochemical profile characteristic of each.

Author Response

Please find herein our point-to-point response. Thank you very much for your suggestions.

This manuscript is well-written and provides a reasonable differential and discussion regarding the occasion when a clear cell neoplasm is encountered in the jaws. The authors have endeavored to help the reader distinguish amongst the various histopathologic possibilities in this regard, and the suggested histochemical and immunohistochemical studies are appropriate.

Should the manuscript move forward with publication, the authors should revise the piece referring to the tumor of interest as "clear cell renal cell carcinoma" rather than "cell clear" as the word-order "cell clear" does not follow convention.

R: We are sorry, the correct definition of the tumor certainly is clear cell renal cell carcinoma and we have corrected the whole paper accordingly

Minor editing and spell-check is requisite including:

line 46 "cloth" should be "clot"

R: corrected as indicated (Pag. 2, line 6)

line 60 should begin "The aim . . .,"

R: corrected as indicated (Pag. 2, line 20)

discussion: paragraph 4 line 4 should read "site of primary tumor"

R: corrected as indicated (Pag. 10, line 17)

discussion: spelling corrected to read "birefringent"

R: corrected as indicated (Pag. 11, line 15)

Finally, the manuscript would be significantly enhanced by the inclusion of a table that endeavors to comprehensively list each of the clear cell lesions one may encounter in the head and neck together with the relevant histochemical and immunohistochemical profile characteristic of each.

R: Such a table was already included in the original version of the manuscript (Table 3, Pag. 9)